# NABLA: NEIGHBORHOOD ADAPTIVE BLOCK-LEVEL ATTENTION FOR EFFICIENT VIDEO GENERATION

## ABSTRACT

Full self-attention in video diffusion transformers scales quadratically with the spatio-temporal token count, making processing the high-resolution clips prohibitively slow and memory-heavy. We introduce **NABLA** (Figure 1), a Neighborhood-Adaptive Block-Level Attention mechanism that builds a per-head sparse mask in three steps: (i) average-pool queries and keys into $N \times N$ blocks, (ii) keep the highest-probability blocks via a cumulative-density threshold, and (iii) optionally union the result with Sliding-Tile Attention (STA) to suppress border artefacts. NABLA drops straight into PyTorch's FlexAttention with no custom kernels or extra losses. On the Wan 2.1 14B text-to-video model at 720p, NABLA accelerates training and inference by up to $2.7\times$ while matching CLIP $(42.06 \rightarrow 42.08)$, VBench $(83.16 \rightarrow 83.17)$ and FVD $(68.9 \rightarrow 67.5)$ scores. During pre-training of a 2B DiT at $512^2$, iteration time falls from 10.9s to 7.5s $(1.46\times)$ with lower validation loss.

## 1 INTRODUCTION

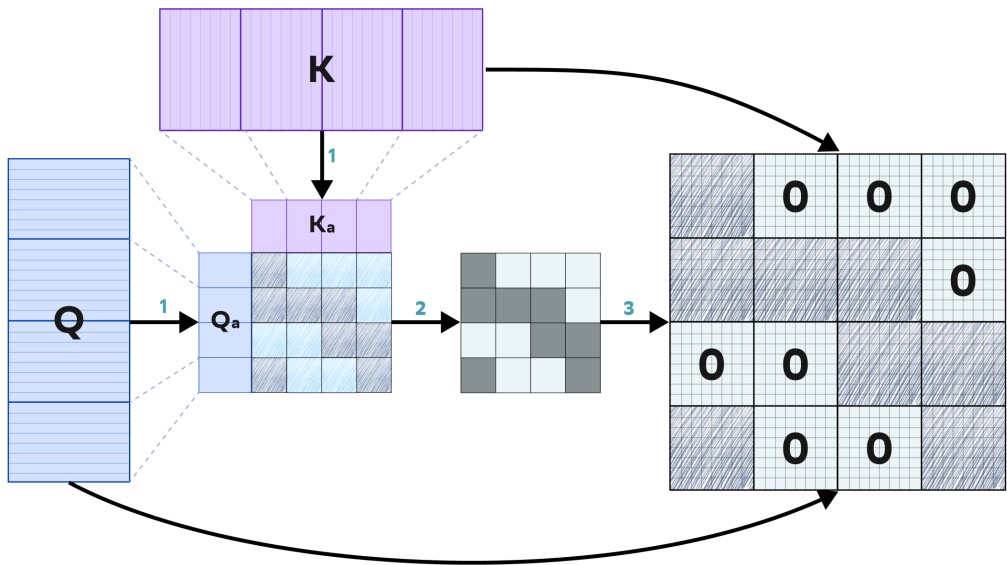

Figure 1: The block-sparse attention mask is computed by (1) reducing the dimensionality of queries (Q) and keys (K), (2) sparsifying the softmax distribution via a cumulative density function (CDF) threshold and binarizing the result, and (3) mapping the sparse mask back to the original input blocks.

Among various generative methods, diffusion Ho et al. (2020) is currently the state-of-the-art approach for generating media content such as images and videos. One of the major milestones in the development of this approach was the introduction of the diffusion transformer. It was first

proposed for image generation in Peebles & Xie (2022), where the authors demonstrated that this architecture surpasses the previously dominant U-Net frameworks for this task and also highlighted its scalability.

Another key advancement in modern content generation using diffusion methods is latent diffusion Rombach et al. (2021), where the diffusion process operates not on raw images or videos but on their compressed representations obtained through variational autoencoders Kingma & Welling (2014). This is particularly important for video generation due to computational complexity.

Diffusion transformers can be broadly categorized into two main classes: CrossDiT and MMDiT Esser et al. (2024). The key differences between these architectures lie in their handling of text embeddings and the attention Vaswani et al. (2017) mechanisms employed. In CrossDiT, text tokens are processed separately from visual tokens and incorporated via cross-attention. In contrast, MMDiT processes text and visual tokens in parallel and blends them through self-attention. This architectural distinction may influence the structure of the attention matrices learned by diffusion transformers.

The full potential of transformers in latent diffusion for video generation was first demonstrated by the closed-source solution Sora OpenAI (2024). Since then, numerous popular closed-source and open-source solutions have emerged, such as MovieGen Polyak et al. (2025), HunyuanVideo Kong et al. (2025), CogVideoX Yang et al. (2025), Kling AI (2024), WAN Wan et al. (2025) and Kandinsky Arkhipkin et al. (2025; 2024). While these models advance video generation capabilities, they share a critical limitation: computationally expensive full attention mechanisms. However recent theoretical analyses Deng et al. (2025) and experimental evidence Tan et al. (2024); Xia et al. (2025); Zhang et al. (2025a); Jiang et al. (2024) have revealed that attention matrices demonstrate inherent sparsity patterns, suggesting significant potential for optimizing computational efficiency through sparse attention mechanisms.

The methods for simplifying attention mechanisms in video generation models presented in the literature can be divided by two main features: dynamism and usage scenario. The first feature determines how important patterns are distinguished in attention masks statically or dynamically. The second feature implies whether the proposed method can be used in zero-shot mode or the model should be trained with it.

The most important and oldest static patterns in attention masks include sliding window attention Beltagy et al. (2020) in NLP tasks and window-based attention (SWIN) Liu et al. (2021) in CV tasks. In recent years, neighbor attention (NATTEN) Hassani et al. (2023); Hassani & Shi (2022) as well as its effective implementations Hassani et al. (2024) has emerged as the successor to SWIN attention. It uses close idea but with intersecting windows. Recent work Sliding Tile Attention Zhang et al. (2025b) builds upon NATTEN principles but optimizes it for efficient computations on GPU using correct size of visual block.

However, although static patterns use empirical knowledge about the data structure in various domains such as NLP or CV, they do not always correspond to the patterns that real transformers obtain during training. Moreover, many works Tan et al. (2024); Xia et al. (2025); Xi et al. (2025); Jiang et al. (2024); Wen et al. (2025) have shown that attention masks differ in unequal blocks, heads, text prompts and even at different steps of video generation. For example, in the MInference Jiang et al. (2024) the authors identify several groups of attention masks for LLM models and select parameters for them on the inference. Sparse VideoGen Xi et al. (2025) divides all attention heads in DiT into spatial and temporal, employing online profiling to dynamically select appropriate patterns. However, these methods consider only fixed number of attention patterns while in the real attention maps there are much more of them. AdaSpa Xia et al. (2025) utilizes dynamic online search to identify suitable block attention masks, employing a hierarchical selection process to determine the necessary sparsity level. SpargeAttn Zhang et al. (2025a) approximates attention masks on compressed queries and keys.

While the aforementioned works primarily focus on accelerating inference for pre-trained video generation models, similar concepts have also been applied to model pre-training and fine-tuning. Numerous studies have explored sparse or simplified attention in NLP tasks Pagliardini et al. (2023); Kitaev et al. (2020); Wang et al. (2020); Gonçalves et al. (2025); Willette et al. (2025). In Native Sparse Attention Yuan et al. (2025) the authors reduce the number of keys and values in LLM in various ways while maintaining the size of the query. Three main patterns stand out: compression,

selection and sliding window, and all of which are applicable to both inference and training. An important work on training a video generator with sparse attention is DSV Tan et al. (2025). It employs a two-stage training. At the first, a low-rank predictor for the attention matrix and a sparsity estimator are trained. At the second stage, the model is trained with the predictor and estimator held fixed.

In summary, numerous methods exist for accelerating attention mechanisms within transformer models. This area has been particularly well-explored in the field of Natural Language Processing. For video generation, this area is still developing. Although several approaches focus exclusively on inference or, at best, fine-tuning, video generation models pretrained with sparse attention remain limited.

In this work, we collect the best practices: simplicity and strong prior masks of Sliding Tile Attention (STA) Zhang et al. (2025b) approach and flexibility of training-based approaches. We present **NABLA**, a Neighbourhood Adaptive Block-Level Attention that utilizes a simple downsampling approach instead of additional training. NABLA dynamically fits the sparsity mask by thresholding the cumulative distribution function for precise attention calculation. Our key contributions include:

- **Efficient content-aware mask construction**, outperforming fixed sparse patterns like STA (validated experimentally).
- **Complementarity with STA** and other acceleration techniques.
- **Simple implementation** via FlexAttention without custom CUDA kernels.
- **Acceleration of both inference and training** of DiT model due to the fast online algorithm.

Extensive evaluations on video datasets demonstrate that NABLA maintains equivalent to full attention VBench, CLIP, FVD and human evaluation scores, achieving up to $2.7\times$ speed-up without excessive additional overhead to the training and inference pipelines.

## 2 BACKGROUND

### 2.1 ATTENTION MECHANISM IN VISUAL DOMAIN

The classical self-attention mechanism, introduced in Vaswani et al. (2017), revolutionized deep learning by enabling dynamic focus on relevant parts of input data through pairwise token interactions. In visual domains, this mechanism processes images and videos by first dividing them into patches and projecting them into an embedding space Dosovitskiy et al. (2021). For an input sequence $X \in \mathbb{R}^{S \times D}$, where $S$ is the number of tokens (e.g., image patches or spatio-temporal video blocks) and $D$ is the embedding dimension, the self-attention mechanism projects these tokens into queries $Q$, keys $K$, and values $V$ using learnable weight matrices $W_Q, W_K, W_V \in \mathbb{R}^{D \times D}$:

$$Q = XW_Q, \quad K = XW_K, \quad V = XW_V. \tag{1}$$

Attention scores are computed using a scaled dot product between queries and keys, followed by a softmax operation to produce the attention matrix $A \in \mathbb{R}^{S \times S}$.

$$A = \text{softmax}\left(\frac{QK^T}{\sqrt{D}}\right), \tag{2}$$

where each entry $A_{ij}$ determines how much token $j$ influences token $i$. The final output is a weighted sum of values based on these scores:

$$\text{Output} = AV. \tag{3}$$

For video diffusion transformers (DiTs) Peebles & Xie (2022), the input consists of spatio-temporal tokens (e.g., $X \in \mathbb{R}^{T \times H \times W \times D}$, where $T$ is the number of frames and $H, W$ are spatial dimensions of the latent space). In this case, the classical self-attention approach faces significant challenges due to the quadratic complexity $\mathcal{O}((T \cdot H \cdot W)^2)$ relative to the number of tokens. High-resolution or long-duration videos exacerbate this issue, as the sequence length grows cubically with spatial and temporal dimensions. In addition, many elements of attention map are near zero due to locality in space and time, which leads to redundant computations. Figure 2 contains examples of attention

maps in which computations on only significant elements have less complexity: $\mathcal{O}(T \cdot H \cdot W)$ ( 2a), $\mathcal{O}(T^2 \cdot H \cdot W)$ ( 2b, 2h), $\mathcal{O}(T \cdot (H \cdot W)^2)$ ( 2d, 2f, 2g). Although full attention theoretically preserves global coherence, its computational cost becomes prohibitive, creating a critical bottleneck for practical applications.

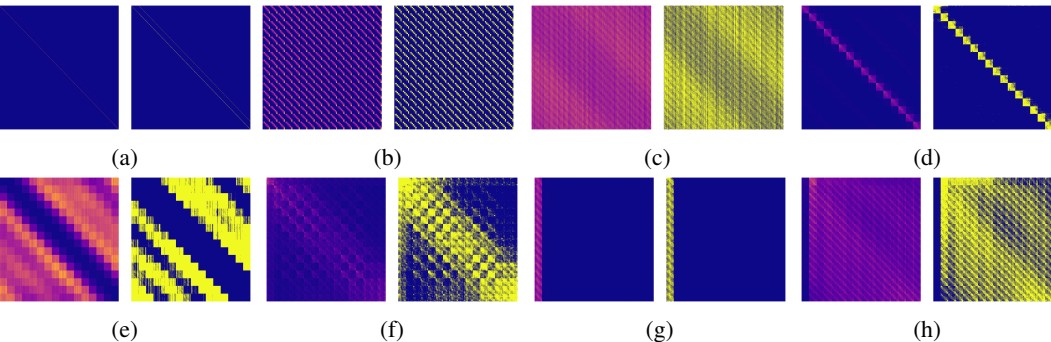

(a)    (b)    (c)    (d)

(e)    (f)    (g)    (h)

Figure 2: Examples of attention maps (left) and corresponding NABLA masks (right) for different heads of Wan 2.1 14B T2I layers.

## 2.2 SLIDING TILE ATTENTION (STA)

To address the limitations of full attention in video generation models, Sliding Tile Attention (STA) Zhang et al. (2025b) was proposed as a hardware-efficient alternative that leverages the inherent 3D locality observed in pretrained video DiTs. STA organizes an input video latent $X \in \mathbb{R}^{T \times H \times W}$ (flattened into $S = THW$ tokens) through a tiling mechanism. Tokens are partitioned into non-overlapping tiles of size $(B_T, B_H, B_W)$, which matches GPU block sizes (for FlashAttention compatibility). STA computes attention only between query tiles and key tiles within a fixed 3D window of size $(W_T, W_H, W_W)$ centered on the query tile:

$$A = \text{softmax}(\frac{QK^\top}{\sqrt{D}} + M), \quad \text{Output} = AV, \tag{4}$$

where $Q, K, V \in \mathbb{R}^{S \times D}$ and $M \in \{-\infty, 0\}^{S \times S}$ is a sparse mask managed implicitly by tile-based sliding. STA ensures $M$ only activates dense blocks, reducing redundant computation. This design ensures queries within a tile attend only to keys in predefined windows, eliminating irregular memory access patterns of traditional sliding window approaches. The number of dense blocks is:

$$S_{\text{dense}} = \left( \frac{W_T}{B_T} \times \frac{W_H}{B_H} \times \frac{W_W}{B_W} \right) \times \left( \frac{T}{B_T} \times \frac{H}{B_H} \times \frac{W}{B_W} \right). \tag{5}$$

By aligning tile sizes with GPU thread blocks, STA minimizes masking overhead and maximizes hardware utilization, achieving up to $10.45\times$ speedup over full attention while maintaining generation quality. A key innovation of STA is its adaptability to head specialization, where different attention heads focus on varying spatial-temporal scales. Through profiling, STA automatically configures optimal window sizes per head, balancing computational efficiency with expressive power. However, STA relies on static window partitioning, which may not fully capture dynamic content-specific patterns, and requires careful tuning to avoid visual artifacts such as blocky boundaries.

## 2.3 MOTIVATION FOR NABLA

During our experiments with STA, we identified a serious issue of object duplication in high-resolution generation and long video sequences (see the examples 6, 7 in Appendix B). This artifact appears in the case of non-optimal STA configuration and is caused by insufficient global attention coverage in the STA approach. Finding the optimal configuration takes a long time, and as we show later, the optimal configuration does not always exist. To address this, we hypothesized that an effective sparse attention algorithm must preserve long-range dependencies—connections between tokens distant in space or time. However, the semantics of such dependencies are inherently complex, making them infeasible to capture with fixed sparsity patterns. While STA provides

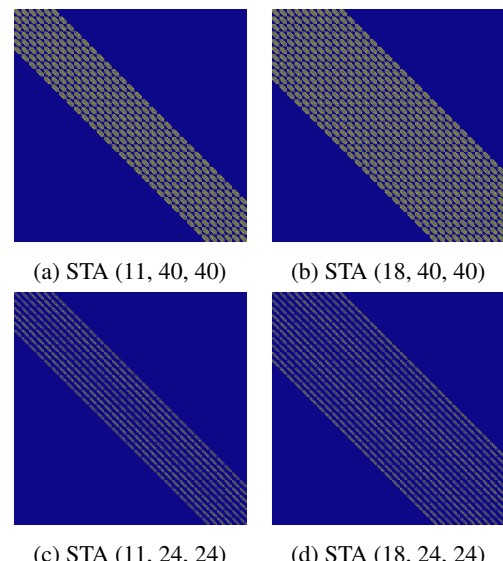

(a) STA (11, 40, 40)    (b) STA (18, 40, 40)

(c) STA (11, 24, 24)    (d) STA (18, 24, 24)

Figure 3: STA masks with different window sizes.

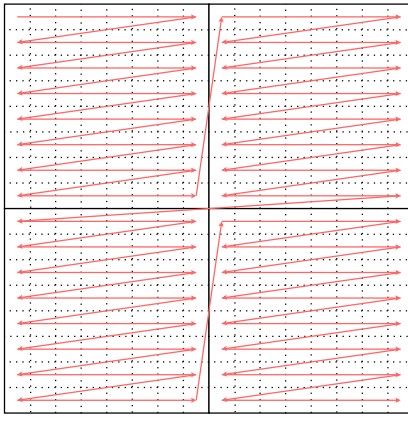

Figure 4: Token reordering illustration for a latent image with height 16, width 16, and patch size 8. The diagram shows how spatial tokens are reorganized into fractal-flattened sequences while preserving their semantic relationships.

significant efficiency gains through hardware-aware sparsity, its static nature limits adaptability to diverse video content. Figure 2 contains examples (2b - 2h) of attention weights that are significantly different from STA pattern (Figure 3).

We argue that adaptive sparsity is essential: the algorithm must dynamically select sparse connections based on the actual input context. This motivates our proposed method, NABLA, which learns context-aware sparsity patterns to maintain global coherence. NABLA utilizes downsampled full attention calculation to dynamically select the most valuable blocks separately for each attention head.

From other side we found, that only adaptive sparsity provides another type of artifacts – visible borders between areas recovered from neighbouring latent pixels. See examples 8, 9 in Appendix B. To address this issue we combine NABLA with STA. This combination provides best results with fixed attention sparsity. As a result, our final approach achieves both computational efficiency and high-quality generation, addressing the limitations of purely static or dynamic approaches.

## 3 METHOD

### 3.1 TRAINING ALGORITHM

This section presents a training (or fine-tuning) version of NABLA algorithm. The method consists of the following parts:

- Token reordering to place tokens of the same spatial block to a continuous sequence
- NABLA adaptive sparsification method that dynamically selects blocks of the feature map for which we perform attention computation
- Sliding Tile Attention (STA) to improve fine-grained quality of the generated videos

### 3.1.1 TOKEN REORDERING

Following the approach of STA Zhang et al. (2025b), we find token reordering crucial for establishing semantic connections between adjacent tokens. Our method employs fractal flattening with spatial patches of size $P \times P$, which groups all tokens within each patch into a contiguous sequence of $P^2$ tokens. Notably, we preserve the original ordering along the temporal dimension.

Figure 4 illustrates this transformation. We apply the reordering operation at the input stage of the DiT network and its inverse at the output stage, ensuring proper spatial relationships while maintaining computational efficiency.

### 3.1.2 NABLA MASK COMPUTATION ALGORITHM

Algorithm 1 presents the NABLA mask computation for Multi-Head Self-Attention. The algorithm takes as input a data sample represented by queries $Q$ and keys $K$, each containing $S$ tokens of dimension $D$ (we omit the batch dimension for simplicity). We denote the number of transformer heads as $h$ to distinguish it from the latent frame height $H$.

The core idea of our method involves computing a full attention map for downsampled versions of $Q$ and $K$, followed by binarization with minimal information loss. The downsampling is performed through average pooling of tokens in blocks of size $N = P^2$, making the reduced attention map computation $N^2$ times more efficient than computing the full attention map.

After computing the reduced attention map, we apply the softmax operation and compute the cumulative distribution function (CDF) for each row. We then binarize the map by retaining only values whose CDF exceeds the threshold $1 - thr$, where $thr$ is algorithm parameter. The binarization yields a unique sparsity pattern for each head, represented by an $S/N \times S/N$ matrix of binary values indicating whether to compute attention for the corresponding $N \times N$ block. Note that the $\mathrm{softmax}$, $\mathrm{sort}$, and $\mathrm{cumsum}$ operations are applied along the last dimension of the input tensor.

---

**Algorithm 1** NABLA Sparse Mask Generation

---

**Require:** Query tensor: $Q \in \mathbb{R}^{h \times S \times D}$, Key tensor: $K \in \mathbb{R}^{h \times S \times D}$, binarization threshold: $thr$, block size: $N$

      **Reduced Attention Map Computation:**
1: $Q \leftarrow \mathrm{reshape}(Q, [h, S/N, N, D])$
2: $K \leftarrow \mathrm{reshape}(K, [h, S/N, N, D])$
3: $Q_a \leftarrow \mathrm{mean}(Q, \dim = -2)$             ▷ Block averaging, $Q_a \in \mathbb{R}^{h \times S/N \times D}$
4: $K_a \leftarrow \mathrm{mean}(K, \dim = -2)$             ▷ Block averaging, $K_a \in \mathbb{R}^{h \times S/N \times D}$
5: $K_a^T \leftarrow K_a.\mathrm{transpose}(-2, -1)$          ▷ $K_a^T \in \mathbb{R}^{h \times D \times S/N}$
6: $A \leftarrow \mathrm{softmax}(\frac{Q_a K_a^T}{\sqrt{D}})$        ▷ Reduced attention map, $A \in \mathbb{R}^{h \times S/N \times S/N}$
      **Binarization via CDF:**
7: $vals, order \leftarrow \mathrm{sort}(A)$                  ▷ Row-wise sorting
8: $cvals \leftarrow \mathrm{cumsum}(vals)$               ▷ Cumulative sum
9: $M \leftarrow cvals \geq 1 - thr$           ▷ Binarization of ordered values
10: $M_\nabla \leftarrow \mathrm{reorder}(M, order)$          ▷ Original order restoration
11: **return** $M_\nabla$

---

### 3.1.3 JOINT NABLA AND STA SPARSITY MASK

We find that combining NABLA with STA results in the best visual quality, benefiting from both our method's adaptive nature and STA's strong prior mask. The STA mask is computed as $M_{STA} = \mathrm{STA\_mask}(T, H, W, W_T, W_H, W_W)$, where:

- $T$: Number of latent frames in the video sample ($T = 1$ for images)

- $H, W$: Latent frame height and width

- $W_T, W_H, W_W$: STA window parameters

The final mask for each data sample is given by $M = M_\nabla \vee M_{STA}$. After the mask is computed, it can be used directly in the Flex Attention Dong et al. (2024) algorithm to improve the training efficiency.

| Metric | Wan2.1-14B | STA (18,24,24) | NABLA (0.4) | NABLA(0.2)+ STA(11,24,24) |
|---|---|---|---|---|
| Subject consistency | 94.6 | 95.00 | **95.01** | 93.18 |
| Background consistency | 98.63 | 98.44 | **98.67** | 98.25 |
| Aesthetic quality | 67.27 | 67.51 | 67.10 | **67.63** |
| Imaging quality | **66.46** | 66.18 | 66.35 | 66.28 |
| Object class | 81.09 | 82.12 | **85.83** | 82.91 |
| Multiple objects | **70.57** | 50.53 | 66.23 | 67.98 |
| Color | 89.83 | 85.61 | 85.19 | **92.22** |
| Spatial relationship | 70.97 | 66.45 | **75.44** | 70.07 |
| Scene | 45.36 | 48.11 | 50.07 | **51.38** |
| Temporal style | 23.34 | 22.95 | **23.46** | 22.99 |
| Overall consistency | 25.80 | **26.65** | 26.10 | 26.00 |
| Human action | **95** | 0.9 | 91 | 93 |
| Temporal flickering | **98.91** | 98.83 | 98.88 | 98.78 |
| Motion smoothness | 98.38 | **98.65** | 98.53 | 98.58 |
| Dynamic degree | 70.83 | 68.06 | 68.05 | **72.22** |
| Appearance style | 22.69 | 22.51 | **23.18** | 23.14 |
| Quality score | **85.15** | 85.05 | 85.02 | 85.03 |
| Semantic score | 75.23 | 71.73 | 75.76 | **76.04** |
| Total score | 83.16 | 82.39 | 83.17 | **83.22** |

Table 1: VBench results for 90% sparsity. Bold values indicate the best performance in each category.

## 4 EXPERIMENTS

### 4.1 FINE-TUNING EXPERIMENTS

We evaluate our method in the fine-tuning setup using the Wan 2.1 14B T2V model Wan et al. (2025) at 720p resolution, focusing specifically on self-attention blocks due to their dominant contribution to overall FLOPs. We implement NABLA alongside STA as our baseline sparse attention method. For reproducibility, we use Flex Attention implementation from PyTorch 2.7. All experiments maintain consistent hardware and software configurations to ensure fair comparisons.

We perform knowledge distillation of the teacher model Wan2.1 T2V 14B using MSE loss. We initialize the student model with the baseline model weights and replace all self-attention blocks with sparse attention. Complete hyperparameter details are provided in Appendix A. Tables 2 and 3 present our key findings. Table 2 presents the sparsity and acceleration settings for STA, NABLA, and STA+NABLA, which are used in subsequent quality evaluations. Detailed evaluation results are provided in Table 1. Key observations from our experiments include:

- NABLA achieves full quality recovery in generation metrics (CLIP, FVD and VBench scores)
- The STA-only configuration shows degradation in VBench semantic scores
- Detailed results (Table 1) reveal STA's particular challenges with multiple objects and spatial relationships
- Pure NABLA and NABLA+STA combinations maintain baseline-level performance across all objective metrics

### 4.2 HUMAN EVALUATION

We conducted a side-by-side human evaluation comparing generated video quality obtained by the baseline model and the finetuned models across various configurations. 50 participants evaluated 20 video pairs each comparing videos on three key perceptual dimensions (prompt alignment, visual quality, dynamics) selecting one option per dimension: left is better, right is better, both are good, both are bad. Results in Table 4 show NABLA's perceptual parity with baseline even at high sparsity.

| Method | Sparsity, % | Inference time, min |
|---|---|---|
| Baseline | 0 | 8.35 |
| STA(18,40,40) | 79.45 | 4.00 |
| NABLA(0.7) | 80.13 | 4.02 |
| NABLA(0.5)+STA(11,40,40) | 81 | 3.58 |
| STA(18,24,24) | 91.28 | 3.08 |
| NABLA(0.4) | 92.5 | 3.07 |
| NABLA(0.2)+STA(11,24,24) | 92.27 | 3.13 |

Table 2: Computational efficiency comparison. All measurements performed on $4\times$ H100 GPUs.

**Evaluation protocol:**

- **Video pairs:** Two 5-second preliminary generated videos for comparing methods are shown for randomly selected prompt (left/right video methods are shuffled).
- **Prompts:** We use 942 diverse text prompts from VBench. The prompt for current videos is also visible to the user.
- **Judge:** The user can watch the videos in repeat mode with the ability to zoom in/out and pause. The user must decide which video (left or right) is better, or both videos are good or bad.
- **Dimensions:**
  - Visual Quality: Artifact freedom and sharpness
  - Motion Naturalness: Better dynamics, physical plausibility and fluidity
  - Semantic Alignment: Prompt-video consistency

## 4.3 PRETRAINING EXPERIMENTS

To verify the applicability of our method during pretraining, we train a custom DiT-based 2B model in three stages:

1. Text-to-image pretraining at $256\times256$ resolution with full attention.
2. Text-to-video pretraining at $256\times256$ resolution with full attention.
3. Text-to-video pretraining at $512\times512$ resolution:
   - (a) With full attention.
   - (b) With the NABLA method (80% sparsity).

The first stage is conducted from scratch, with each subsequent stage initialized using weights from the previous stage. The first two stages are common to all experiments.

We compare the convergence of training and validation losses between stages 3(a) and 3(b). Figure 5 shows that the NABLA model achieves better convergence than its full attention counterpart.

| Method | CLIP score↑ | FVD score↓ | VBench score↑ Quality | VBench score↑ Semantic | VBench score↑ Total |
|---|---|---|---|---|---|
| Wan2.1-14B (Baseline) | 42.06 | 68.9 | **85.15** | 75.23 | 83.16 |
| STA (18,24,24) | 41.51 | 74.06 | 85.05 | 71.73 | 82.39 |
| NABLA(0.4) | **42.08** | **67.5** | 85.02 | 75.76 | 83.17 |
| NABLA(0.2)+ STA(11,24,24) | 41.98 | 73.78 | 85.03 | **76.04** | **83.22** |

Table 3: Model quality metrics after fine-tuning. NABLA variants maintain comparable performance to baseline even at 90% sparsity.

| Winner | Semantic Alignment (%) | Visual Quality (%) | Motion Naturalness(%) | Overall (%) |
|---|---|---|---|---|
| Baseline better | 19.9 | 31.4 | 10.5 | 20.3 |
| NABLA(0.7) better | 13.3 | 26.7 | 15.2 | 18.4 |
| Both good | 66.7 | 40 | 64.8 | 57.2 |
| Both bad | 0.9 | 1.9 | 9.5 | 4.1 |

Table 4: Human evaluation results (SBS test). NABLA maintains perceptual quality comparable to baseline at 80% sparsity.

Furthermore, each training iteration takes 10.9 seconds for the full attention model compared to 7.5 seconds for NABLA, resulting in a $1.46\times$ speedup.

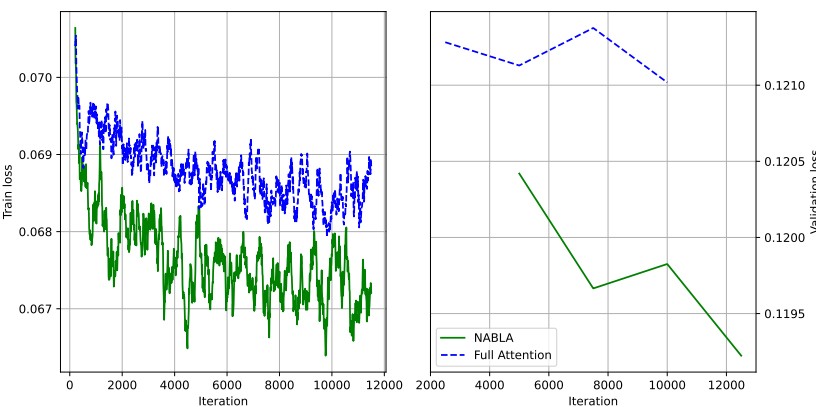

Figure 5: Training convergence for full attention and NABLA models at $512\times512$ resolution. NABLA achieves lower training and validation losses.

## 5 CONCLUSION

By dynamically adapting to sparsity patterns through block-wise attention with adaptive thresholding, our method achieves:

- Significant computational efficiency: Up to $2.7\times$ faster training and inference compared to full-attention baselines
- Minimal quality degradation: Near-identical performance to full attention in quantitative metrics (CLIP, FVD, VBench) and human evaluations
- Hardware-agnostic implementation: Seamless integration with PyTorch's Flex Attention without custom CUDA kernels

Extensive experiments demonstrate NABLA's superiority over static sparsity approaches, particularly in preserving long-range dependencies and handling complex spatial-temporal relationships. Our hybrid approach combining NABLA with static patterns further enhances visual quality by mitigating boundary artifacts while maintaining efficiency.

The proposed approach establishes a new state-of-the-art for efficient video generation, enabling high-resolution synthesis with reduced computational demands.

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

# APPENDICES

## A FINE-TUNING HYPERPARAMETERS

We performed knowledge distillation on the full-attention Wan2.1 open-source model using MSE loss. The training utilized a specially curated dataset of high-quality, high-dynamic videos and high-quality images. The experiments were conducted on 256 H100 GPUs, using the following hyperparameters:

- Total batch size: 64
- Sequence parallel: 4
- Training steps: 1600
- Optimizer: AdamW with:
  - Learning rate: 1e-6
  - Weight decay: None
  - Betas: (0.9, 0.95)
  - Epsilon: 1e-8
- Gradient norm: 0.01

## B  GENERATION EXAMPLES

The following examples (Fig. 6,7,8,9) demonstrate generation results from different model configurations, presented clockwise from the top-left corner: Full Attention (Baseline Wan 2.1), STA(18,24,24), NABLA(0.4), and NABLA(0.2) + STA(11,24,24).

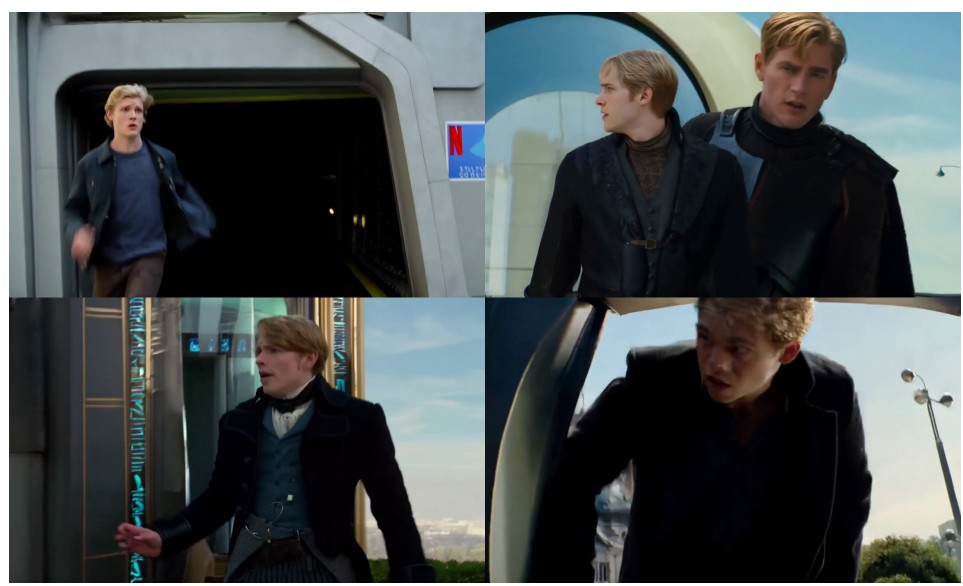

Figure 6: Input prompt: *Movie scene of a time portal opening up in a modern city, a 18th century young blonde man walks out of it looking confused, close-up, sci-fi, Netflix Original, professionally color graded, 35mm film*

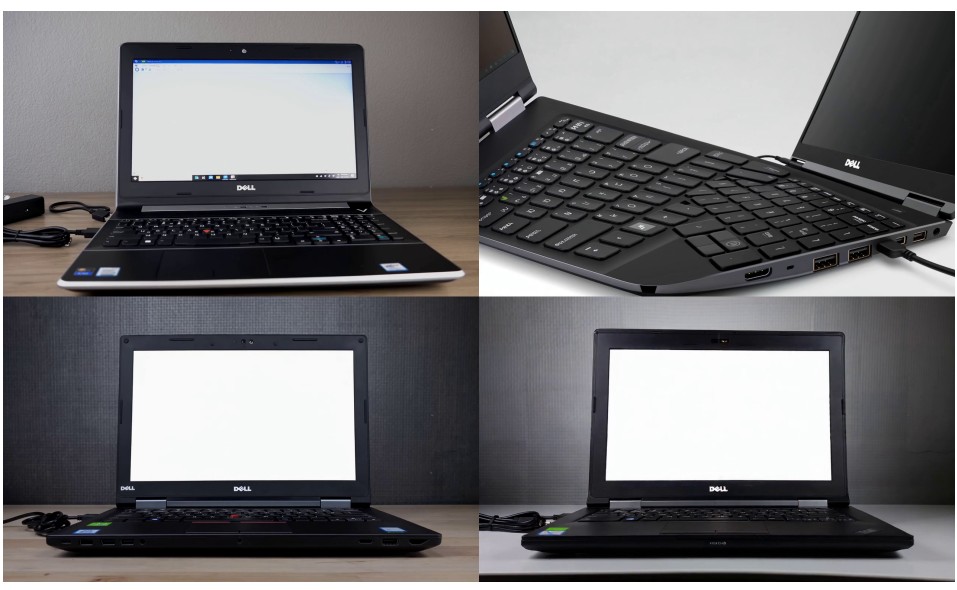

Figure 7: Input prompt: *A sleek black laptop made of durable aluminium with a flat rectangular shape. It is a medium-sized device with a 14-inch screen. The laptop features a backlit keyboard and comes with a charger. The text on the device reads 'Dell.'*

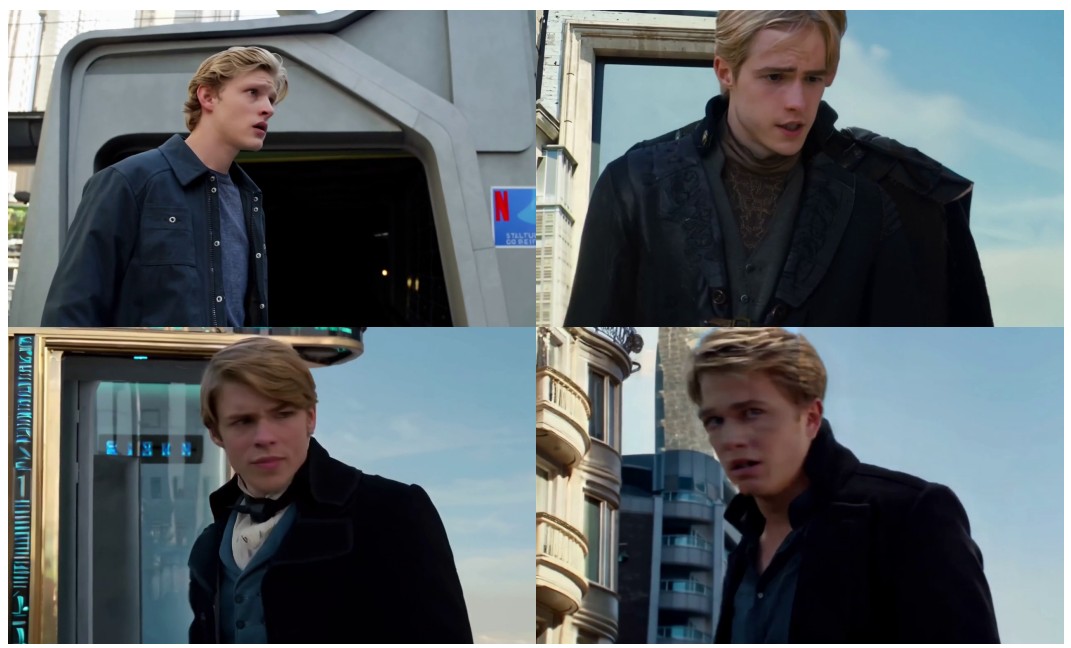

Figure 8: Input prompt: *Movie scene of a time portal opening up in a modern city, a 18th century young blonde man walks out of it looking confused, close-up, sci-fi, Netflix Original, professionally color graded, 35mm film*

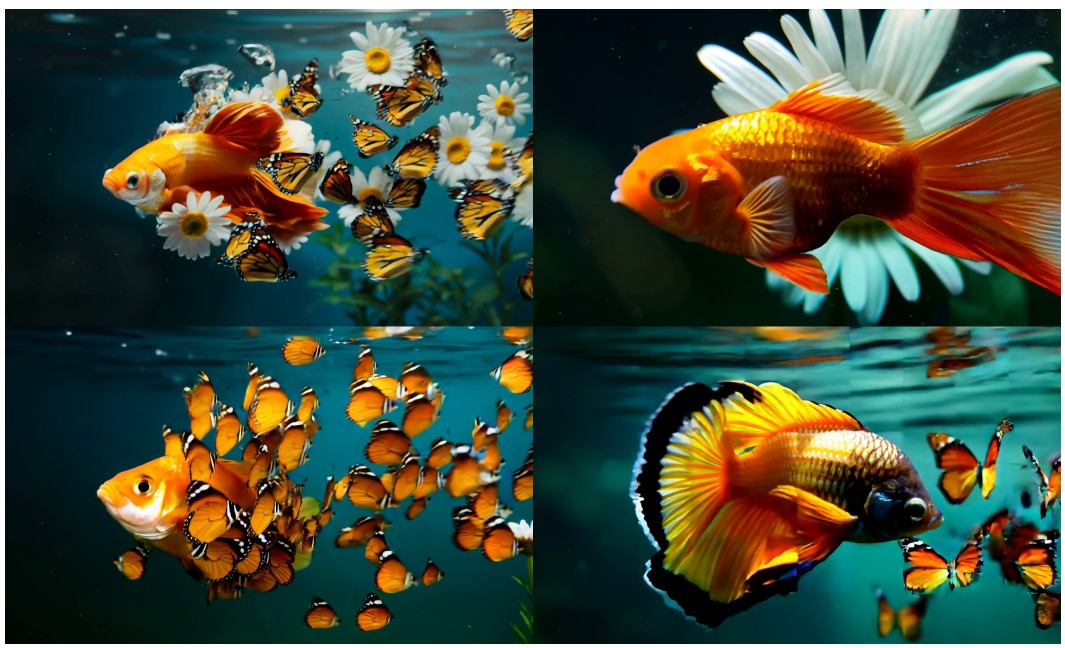

Figure 9: Input prompt: *"Cinematic shot of a Beta fish swimming, moving dynamically in the water. A daisy is transformed into a group of butterflies. The fish has orange, blue and yellow colors. Beautiful nature documentary, low contrast, 35mm, color correction.*

