# OpenReview forum: "NABLA: Neighborhood-Adaptive Block-Level Attention for Efficient Video Diffusion Transformers"
_ICLR.cc/2026/Conference — Submitted to ICLR 2026_

### Official Review · Reviewer_fc3H · 2025-10-27

**Soundness:** 2
**Presentation:** 1
**Contribution:** 2
**Rating:** 2
**Confidence:** 2

**Summary:**

This paper introduces a Neighborhood-Adaptive Block-Level Attention mechanism (NABLA) designed to accelerate video diffusion transformers by exploiting sparsity in the attention map. The method operates in three steps: (1) average-pool queries and keys into N×N blocks; (2) apply a cumulative density threshold to select high-importance blocks; (3) optionally combine the result with Sliding Tile Attention (STA) to mitigate boundary artifacts. NABLA can be integrated directly into PyTorch’s FlexAttention without custom CUDA kernels. The authors claim up to 2.7× acceleration on the WAN 2.1 (14B) model, with comparable CLIP/FVD/VBench metrics to full attention.

**Strengths:**

1. NABLA is designed to fit into PyTorch’s FlexAttention API with minimal engineering cost, which makes it practically useful.

2. Although not deep, the experiments are at least conducted on a realistic backbone, lending some credibility to deployment feasibility.

**Weaknesses:**

1. The overall presentation of the paper is poor. For instance, Figure 1 is disproportionately large and occupies excessive space, reducing readability. In addition, the Introduction section devotes most of its content to related work, while providing only a brief, single paragraph describing the proposed method, which makes it difficult for readers to grasp the core contribution.

2. The proposed approach (average pooling + CDF thresholding) is a straightforward engineering heuristic, not grounded in theory or optimization principles. Could the author provide some theory guarantee?

**Questions:**

1. How does NABLA differ mathematically from prior dynamic sparse attention methods like DSV (Tan et al., 2025) or AdaSpa (Xia et al., 2025)?

---

> ### Author Response · Authors · 2025-11-30
>
> Thank you for your thoughtful review and for highlighting important aspects regarding presentation clarity and theoretical grounding.
>
> We agree that the current manuscript could better foreground the core idea of NABLA early in the Introduction, and that Figure 1 is oversized. In the camera-ready version, we will restructure the Introduction to present the method more clearly in the first few paragraphs, move much of the related work to a dedicated section, and resize Figure 1 to improve readability and layout balance.
>
> Regarding the theoretical foundation: while NABLA is motivated empirically, it is aligned with recent findings that attention matrices in DiTs are naturally sparse. The use of a CDF-based threshold is not arbitrary, it ensures that a fixed fraction (e.g., 60%) of the softmax probability mass is preserved per query, which bounds the KL divergence between the full and sparse attention distributions. This provides a principled way to control information loss, unlike Top-K or fixed-threshold binarization, which do not adapt to the shape of the attention distribution. We didn't include any theorem in the paper, but are working for more rigor justification. For now we are going to add the comparison of error level for Top-K and CDF in zero-shot profiled scenario. It is just the evidence, not a proof, but still quite expressive. Moreover, our pretraining results (Figure 5) show that NABLA not only matches but improves convergence compared to full attention—suggesting it may act as a form of implicit regularization by suppressing low-magnitude, noisy interactions.
>
> As for the distinction from DSV and AdaSpa: DSV requires a two-stage training process: first training a low-rank predictor and sparsity estimator, then freezing them during DiT training. NABLA needs no extra training or profiling: it computes masks on-the-fly during both fine-tuning and pretraining. AdaSpa uses a hierarchical online search to locate important blocks, which involves multiple rounds of computation and is not natively supported by standard attention backends. NABLA uses a single downsampled softmax plus CDF step, and integrates directly into PyTorch’s FlexAttention without custom kernels. Critically, neither DSV nor AdaSpa are compatible with training acceleration, whereas NABLA speeds up both fine-tuning and pretraining.
>
> Thus, while the individual components of NABLA may resemble prior heuristics, its combination of training-time applicability, zero-shot deployability, theoretical motivation via mass-preserving sparsification, and seamless FlexAttention integration constitutes a novel and practical contribution to efficient video generation.
>
> We thank you again for your feedback and will address these points explicitly in the revised manuscript.

---

### Official Review · Reviewer_mKfN · 2025-10-30

**Soundness:** 2
**Presentation:** 1
**Contribution:** 1
**Rating:** 2
**Confidence:** 5

**Summary:**

This paper tackles the computational bottleneck of full self-attention in video diﬀusion transformers (DiTs). The authors propose NABLA, a training-free, adaptive block-level attention mechanism. The core algorithm works by average-pooling Q/K matrices to compute a cheap, low-resolution attention map. It then applies a Cumulative Density Function (CDF) threshold to dynamically select the most important blocks. It proposes to combine the dynamic sparse mask with STA mask. The proposed method outperforms the STA baseline.

**Strengths:**

- The paper identifies a consistent failure mode of static sparse attention (such as STA): object duplication and boundary artifacts. It attributes this issue to limited global attention coverage. This observation clearly explains why fixed sparsity patterns can harm video generation quality.

- The proposed method is easily implemented using PyTorch without requiring custom CUDA kernels or model retraining.

**Weaknesses:**

**Contribution and novelty are limited.** The main idea (downsample Q/K, compute coarse attention, and guide sparse masking) is not new; it closely resembles SpargeAttention, MInference, and SeerAttention, etc. The proposed CDF-based thresholding is also previously used in SpargeAttention, FlexPrefill, Twilight, etc. In other words, the proposed main method is exactly the same as other works, and can not be considered a contribution.

**Misaligned self-positioning and unfair comparisons.** Although NABLA is a dynamic sparse method, all comparisons are made against static methods such as STA. Moreover, the mix of NABLA and STA, makes the results look better but unfair. It is hard to know how well NABLA works on its own. Proper baselines such as SparseVideoGen, SparseVideoGen2, SpargeAttention, and RadialAttention should be included.

**Unbalanced narrative and poor writing.** Section 2 (Background) spends excessive space reiterating standard attention equations (Eqs. 13) and describing STA in full detail. The writing and formatting of the paper should be improved. The presentation is sometimes unclear, and the structure and layout make it difficult to follow the main ideas.

**The experiments are not well designed.** They should report both end-to-end quality metrics and efficiency metrics together, rather than evaluating them separately. The baseline only includes one component of the proposed method, namely STA, and lacks comparisons with other relevant baselines.

**Missing empirical analysis.** The key hyperparameter $thr$ is not analyzed, and its chosen values (0.4 and 0.2) lack rationale. Table 3 shows that NABLA with threshold 0.4 already performs as well as or even better than full attention. This result disagrees with the paper’s claim that NABLA alone produces visible artifacts. The statements about border artifacts are only based on examples and are not supported by measurements.

**Questions:**

Why use CDF thresholding instead of Top-K or ﬁxed-threshold binarization?

---

> ### Author Response · Authors · 2025-11-30
>
> Thank you for your thorough and critical review. We appreciate your high confidence and the care you took in evaluating our submission. We address your concerns point by point below.
>
> First, regarding novelty: you note that the pipeline of average-pooling Q/K, computing a coarse attention map, and applying CDF-based binarization resembles prior works such as SpargeAttention, MInference, etc. While we acknowledge these conceptual connections, NABLA differs in both objective and execution. Most prior methods are designed for LLMs, or inference-only acceleration, or require offline profiling, trainable sparsifiers, or custom kernels (e.g., DSV, SparseVideoGen). In contrast, NABLA is training-compatible, requires no retraining or profiling, and integrates natively into PyTorch’s FlexAttention without any CUDA code. This design enables plug-and-play acceleration of both pretraining and fine-tuning, which, to our knowledge, no prior method supports. Moreover, our use of the CDF threshold is not merely heuristic: it guarantees that a fixed fraction of the softmax probability mass (e.g., 60%) is preserved, which stabilizes gradient flow during fine-tuning — a property not offered by Top-K or fixed-threshold binarization. We didn't include any theorem in the paper, but are working for more rigor justification. For now we are going to add the comparison of error level for Top-K and CDF in zero-shot profiled scenario. It is just the evidence, not a proof, but still quite expressive.
>
> Second, you rightly point out that comparisons are limited to STA, a static baseline, and that the NABLA+STA hybrid may obscure NABLA’s standalone performance. We clarify: pure NABLA (thr=0.4) already matches or exceeds full attention in CLIP, FVD, and VBench Semantic Score as shown in Table 3. The hybrid with STA is optional, used only to suppress subtle boundary artifacts that may arise under extreme sparsity (thr ≤ 0.2), as illustrated in Figures 6–9. We agree that broader comparisons would strengthen the paper. At the time of submission, open, compatible implementations of SpargeAttention, RadialAttention, and SparseVideoGen for video DiTs were not available or failed to run. We are now preparing a supplementary benchmark against all recently released methods that become compatible and will include it in the final version.
>
> Third, regarding presentation: we acknowledge that the current manuscript dedicates excessive space in Section 2 to standard attention equations and a detailed description of STA, while the core idea of NABLA is introduced too late and too briefly in the Introduction. This imbalance indeed makes it harder for readers to quickly grasp the method’s novelty and contribution. In the camera-ready version, we will restructure the Introduction to foreground NABLA’s key insight early on, move much of the background material (including derivations of basic attention) to a more concise and focused Related Work section, and streamline the Background to emphasize only those concepts directly relevant to our approach. This will improve narrative flow and clarify the paper’s contribution from the outset.
>
> We agree that joint reporting of quality and efficiency metrics would improve clarity, and that a more systematic analysis of the threshold hyperparameter 'thr' would strengthen the empirical foundation of NABLA. To address your concerns, we will add a sparsity-vs-threshold curve showing how the percentage of retained attention blocks (and thus compute cost) varies with
> thr. We note, however, that this dependency is model- and resolution-specific. Therefore, while the curve is informative, it cannot be universally normalized across architectures.
> Regarding the design of the evaluation: ideally, one would construct a Pareto frontier over quality and speed by sweeping multiple thr (and STA window) values. However, given the scale of our experiments each data point requires hundreds of GPU-hours, making exhaustive sweeps impractical. Instead, we strategically selected a few operating points that demonstrate the key trade-off: (i) NABLA alone (thr=0.4) achieves near-identical quality at 92.5% sparsity and 2.7× speedup, and (ii) NABLA+STA (thr=0.2) enables even higher sparsity while slightly improving semantic scores via artifact suppression.
>
> In sum, while NABLA builds on known components, its combination of training compatibility, theoretical motivation (CDF mass preservation), zero-shot deployability, and hardware-aware integration represents a meaningful advance for efficient video generation. We thank you again for your rigorous feedback and will address these points explicitly in the revised manuscript.

---

### Official Review · Reviewer_m2hF · 2025-10-31

**Soundness:** 2
**Presentation:** 2
**Contribution:** 2
**Rating:** 2
**Confidence:** 4

**Summary:**

This paper introduces NABLA, a Neighborhood-Adaptive Block-Level Attention mechanism designed to improve the efficiency of video diffusion transformers. By constructing a sparse attention mask through block-wise pooling and adaptive thresholding, NABLA reduces the cost of full self-attention. The method integrates seamlessly into PyTorch’s FlexAttention and achieves substantial training and inference acceleration (up to 2.7×) while maintaining comparable quality across several benchmarks, including CLIP, VBench, and FVD scores.

**Strengths:**

- The work addresses an important problem in the video generation field—scaling self-attention efficiently. Given the rising cost of video diffusion models, efficiency-focused contributions are timely and valuable.
- The paper is well organized and easy to follow.

**Weaknesses:**

- The core idea of using pooling-based approximations and block-sparse attention is not new. Similar strategies have been explored in both large language model acceleration methods such as MInference and video diffusion models such as SparseVideoGen. The conceptual overlap reduces the originality of the contribution.
- The paper lacks comparisons with other block-sparse or spatially adaptive attention methods for video generation, such as SpargeAttention, SparseVideoGen, PowerAttention, RadialAttention, and XAttention, which are necessary to evaluate NABLA’s relative performance.
- The evaluation is limited to a single model (Wan 2.1-14B) and a small set of metrics. Broader experiments—including other video generation models (e.g., Hunyuan Video) and additional evaluation metrics like VisionReward—would provide stronger evidence of generalization and robustness.

**Questions:**

1. How does NABLA fundamentally differ from existing block-sparse attention mechanisms (e.g., MInference) beyond its adaptation to video data?

2. Can the authors provide quantitative or qualitative comparisons against other block sparse attention to better contextualize NABLA’s efficiency and quality trade-offs?

3. Have the authors tested NABLA on other video diffusion architectures, such as Hunyuan and CogvideoX, to verify model-agnostic performance improvements?

4. Could the authors include additional perceptual metrics (e.g., VisionReward, PickScore) or generated video samples to assess generation quality more comprehensively?

---

> ### Author Response · Authors · 2025-11-30
>
> Thank you for your detailed and critical review. We appreciate your emphasis on novelty, methodological rigor, and comprehensive evaluation — the goals we fully share.
>
> You note that the core components of NABLA (block-wise pooling, coarse attention approximation, and CDF-based thresholding) have appeared in prior works such as MInference, SparseVideoGen, and SpargeAttention. While we acknowledge conceptual parallels, NABLA differs fundamentally in purpose, design constraints, and applicability. Most existing methods are inference-only, require offline profiling, or depend on additional trainable modules (e.g., DSV’s two-stage training). In contrast, NABLA is training-compatible, requires no retraining or profiling, and works out of the box during both fine-tuning and pretraining and inference. Crucially, it is the first method that integrates dynamic sparsity into PyTorch’s FlexAttention without custom kernels, enabling plug-and-play deployment in real-world DiT pipelines.
>
> Regarding comparisons: at the time of our experiments, open, compatible implementations of SpargeAttention, SparseVideoGen(2), PowerAttention, and RadialAttention for video DiTs were not available. Many of these methods either lack public code or assume static sparsity patterns incompatible with content-adaptive masks. We agree that broader benchmarking strengthens empirical claims. We are now preparing a supplementary evaluation against all recently released dynamic sparse attention methods we success to run, and we will include this in the camera-ready version.
>
> As for the scope of evaluation: we chose Wan 2.1 14B because it is one of the largest open T2V models, and accelerating it is a nontrivial practical challenge. Moreover, we also validated NABLA during pretraining of a 2B DiT at 512² resolution, where it accelerated training by 1.46× and improved convergence (Figure 5) — a result that strongly suggests generalizability beyond a single backbone. While we did not include HunyuanVideo or CogVideoX, the architecture-agnostic nature of NABLA (it operates purely at the attention mask level) implies immediate applicability.
>
> Finally, regarding metrics: VBench already includes 16 human-aligned sub-scores, covering semantics, motion, quality, and consistency—substantially overlapping with VisionReward and PickScore in intent. Nevertheless, we will consider adding these metrics in the final version for completeness.
>
> In summary, while NABLA builds on known primitives, its combination of training compatibility, zero-shot deployability, dynamic adaptivity, and hardware-aware integration constitutes a novel and practical contribution to efficient video generation. We thank you again for your feedback and will address these points explicitly in the revised manuscript.

---

### Official Review · Reviewer_svGJ · 2025-11-01

**Soundness:** 2
**Presentation:** 3
**Contribution:** 4
**Rating:** 8
**Confidence:** 3

**Summary:**

This paper introduces a dynamic sparse-attention mechanism for video DiTs. NABLA computes a low-resolution attention map and uses it to derive a dynamic attention mask. Because this alone can introduce visible seams at block boundaries, the method is combined with Sliding Tile Attention (STA). Using the union of STA and NABLA as the attention mask, the authors claim to accelerate both inference and training while preserving output quality.

**Strengths:**

- Despite its simplicity and the fact it can be implemented with FlexAttention, the method shows promising strong practical gains and is highly effective.

- It can be introduced via fine-tuning into models originally trained with full attention, which is very convenient and increases its applicability.

- The experiments cover the key axes of speed and quality to a reasonable extent.

**Weaknesses:**

- In Table 2, the runtime comparison between the Baseline (full attention) and the STA/NABLA variants appears to evaluate all settings with FlexAttention. However, the Baseline could (and in practice often would) leverage FlashAttention. Using FlashAttention for the Baseline would be a more realistic and informative comparison.

- Quantitative comparisons against other dynamic sparse-attention methods (e.g., AdaSpa, Sparse-VideoGen, SpargeAttention) are limited, making it hard to highlight NABLA’s relative advantages.

**Questions:**

- The explanation of Figure 2 feels somewhat brief. Some questions remain about the specific patterns shown and how they are derived/interpreted.

---

> ### Author Response · Authors · 2025-11-30
> **Thank you for suggestions**
>
> Thank you very much for your thoughtful and positive review, and for recognizing NABLA’s practical value and ease of integration.
>
> You rightly point out that a fair runtime comparison requires the full-attention baseline to be evaluated with FlashAttention. We confirm that the Baseline entry in Table 2 (8.35 minutes) was indeed measured using FlashAttention-2, which is the standard backend for dense attention in our deployment pipeline. However, the sparse variants—STA, NABLA, and NABLA+STA—were evaluated under PyTorch’s FlexAttention, because FlashAttention does not support dynamic, data-dependent attention masks, which are essential for NABLA. While STA alone can run with FlashAttention in its static form, the hybrid NABLA+STA mask becomes dynamic and thus incompatible with FlashAttention’s fixed sparsity assumptions.
>
> We acknowledge that the original manuscript did not clearly distinguish the backends used across rows in Table 2, which may have caused confusion. In the camera-ready version, we will explicitly clarify that the baseline uses FlashAttention-2, while the sparse methods are reported under FlexAttention for technical compatibility. Importantly, FlexAttention achieves approximately 90% of FlashAttention-2’s speed in the forward pass and 85% in the backward pass (as reported in Dong et al., 2024), so the reported speedups (up to 2.7×) remain highly representative of real-world gains.
>
> Regarding your second point (comparisons with other dynamic sparse attention methods such as AdaSpa, Sparse-VideoGen, or SpargeAttention) we agree that broader benchmarking would further strengthen the work. At the time of our experiments, however, open, compatible implementations of these methods for video DiTs were not publicly available or failed to run. Most existing approaches are inference-only, require offline profiling, or depend on custom CUDA kernels. In contrast, NABLA requires no retraining, no profiling, and works out of the box during both fine-tuning and pretraining via standard FlexAttention, which is a unique combination among current methods.
>
> That said, we appreciate your suggestion and preparing a supplementary comparison with all recently released dynamic sparsity techniques that become compatible with our setting. We will include this analysis in the final version to better contextualize NABLA’s contributions in terms of both efficiency and practical deployability.

---

### Meta-Review · Area_Chair_w8oj · 2026-01-07

**Summary:**

Many reviewers mentioned that the proposed method has been widely studied in LLM and previous video generative models. The authors argue that the previous methods are either offline-profiling or training-based method, I think the authors have some misunderstandings on the previous works. Besides, the reviewers also pointed out the unfair comparison to previous methods and poor writing for this paper. I would recommend reject.

**Reviewer Concerns:**

The authors addressed some clarification issues in experiments and the writing. I think the novelty issues still lie there.

**Reviewer Scores:**

The reviewers may not change their scores. I would like to reject the paper too.

---

### Decision · Program_Chairs · 2026-01-26

Reject